# Linear Attention Optimized GPU Kernel Implementation

## Abstract

The original softmax-based attention mechanism (regular attention) in the extremely successful Transformer architecture computes attention between $N$ tokens, each embedded in a $D$-dimensional head, with a time complexity of $O(N^2D)$. Given the success of Transformers, improving their runtime during both training and inference is a popular research area. One such approach is the introduction of the linear attention (LA) mechanisms, which offers a linear time complexity of $O(ND^2)$ and have demonstrated comparable accuracy to regular attention. However, LA in practice lags behind its theoretical efficiency. We propose a novel method for LA's forward and backward passes, along with a highly-optimized CUDA implementation. Our approach outperforms the state-of-the-art by $3.3\times$ in speed and reduces memory consumption by $3.6\times$. We validate these improvements in both single-layer and end-to-end settings by training a 1.4 billion parameter language model, which demonstrates similar expressivity to regular attention on major reasoning benchmarks.

## 1 Introduction

Transformers are the foundational deep learning architecture behind many recent successes in diverse fields such as natural language processing, speech, computer vision, biology, and more. They incorporate a Softmax-based all-to-all "Attention" mechanism (Regular Attention) between input tokens Vaswani et al. (2017). While this mechanism has proven to be extraordinarily effective in learning tasks, it has a time and memory complexity of $O(N^2D)$ and $O(N^2 + ND)$ respectively, where $N$ is the number of tokens and $D$ the dimension per attention head. Efficient hardware implementation can offer constant speedup, and significant memory reduction, and a notable example is the widely used FlashAttention-2 Dao (2024); Dao et al. (2022), where the memory complexity is reduced to $O(ND)$, but the time complexity remains $O(N^2D)$. With industrial models trending towards increasingly large sequence lengths ($N = 10^7$ in Llama4[1]), the growing ubiquity of Transformers in generative AI, and the significant interest in deploying large language models on computationally limited devices such as smartphones, the search for efficient attention mechanisms is a crucial area of research.

One promising avenue for an efficient attention implementation is the linear attention (LA) mechanism, also known as Kernel Separation. This approach calculates the all-to-all attention with a time complexity of $O(ND^2)$ by employing a linear attention kernel $(a + bx)$ instead of the exponential softmax kernel $(\exp x)$ employed in Regular Attention, and has been shown in previous work to achieve comparable results on various benchmarks. The motivation for LA stemmed from observing the mathematical similarities between Recurrent Neural Networks (RNNs) and Transformers when using a linear attention kernel Katharopoulos et al. (2020). While both RNNs and LA-based Transformers could theoretically yield the same results, RNNs suffer from limited parallelization due to their sequential nature, whereas Transformers are inherently highly parallelizable. Recent work Yang et al. (2023) has explored improving the parallelization of LA within RNNs through I/O-aware implementation. Furthermore, several studies suggest that LA can be computed in linear time using the Transformer architecture itself Han et al. (2023); Cai et al. (2022); Zhang et al. (2024); Wang et al.; Shen et al. (2021) and accelerated with Speculative Decoding You et al. (2024). However, these approaches often underperform their RNN counterparts due to a lack of efficient GPU kernel implementation. Appendix B provides a discussion on the difference of RNN and Transformer-based approaches.

Another line of research proposes combining LA with Regular Attention, where some of the attention layers compute attention over a limited window of $\sqrt{N}$ adjacent tokens, while others apply LA across all tokens Qin

---

[1] https://ai.meta.com/blog/llama-4-multimodal-intelligence/

et al. (2022a; 2023; 2024). Table 1 provides a summary of several approaches. Other approaches aimed at efficient attention include hierarchically-nested attention Ma et al. (2021), the use of $\cos x$ as the attention kernel Qin et al. (2022b), approximating Regular Attention with the fast multipole method Nguyen et al. (2021), attending to a sparse set of tokens per layer Zaheer et al.; Zhou et al., and projecting tokens into a lower-dimensional subspace Yu et al. (2023); Singhania et al. (2024). However, these alternative methods have not yet achieved the same level of practical success as Linear Attention. Further, they also lack the highly optimized GPU implementations that regular attention has, and which we seek to provide for the transformer based linear kernel attention in this paper.

Due to the less established nature of LA, its implementation is far less optimized compared to Regular Attention, and leaving considerable room for improvement. Given LA's apparently comparable expressivity on certain problems and its increasing popularity, evidenced by deployable LA LLMs MiniMax et al. (2025), enhancing LA implementation efficiency is crucial. Our contribution is as follows:

- We derive a novel method for calculating the forward and backward-pass of attention layer with LA based on the Transformer architecture in time complexity of $O(ND^2)$ and $O(ND)$ memory (Section 3). This method aims to enable a parallelized implementation with minimal off-chip memory access.

- We implement the forward and backward-pass on GPU using CUDA, extensively optimizing both data movement and thread scheduling (Section 4).

- We evaluate our implementation in an isolated setting as a standalone attention layer, and in an end-to-end setting where we train a 1.4 billion parameter LLM. We achieve $3.3\times$ speedup and a $3.6\times$ reduction in memory consumption compared to the state-of-the-art LA implementation Yang et al. (2023) on A6000 GPU.

By improving the implementation efficiency of Linear Attention (LA) is critical for its practical adoption, as it directly enables the deployment of large transformer models on resource-constrained platforms like edge devices and mobile hardware. This gain in efficiency democratizes access to advanced AI capabilities by allowing complex models to run effectively on smartphones, embedded systems, and other low-power devices. To elaborate, LA enables faster inference speeds and lower energy consumption, which are essential for real-time applications and battery-powered systems. Moreover, the predictable linear scaling of efficient LA simplifies deployment planning and optimizes batching strategies, making it a viable solution for production environments operating under strict computational budgets.

## 2  Background

At its core, the Transformer architecture Vaswani et al. (2017) processes sequential data through three main components. First, the input sequence is transformed into three sets of matrices Query ($\mathbf{Q}$), Key ($\mathbf{K}$), and Value ($\mathbf{V}$) by passing the input embeddings through linear projection layers. The attention layer then

| Mechanism | Attention Kernel | Time Complexity | Memory Complexity (w. / w.o. causal mask) | Forward-Pass Time (ms) | Forward-Pass Memory (GB) |
|---|---|---|---|---|---|
| Regular Attention | $f(x) = \exp x$ | $O(N^2 D)$ | $O(N^2 + ND)$ / $O(N^2 + ND)$ | OOM | OOM |
| FlashAttention-2 Dao (2024) | $f(x) = \exp x$ | $O(N^2 D)$ | $O(ND)$ / $O(ND)$ | 100 | 1.5 |
| Speculative Decoding LA You et al. (2024) | $f(x) = bx$ | $O(ND^2)$ | $O(ND)$ / $O(ND^2)$ | OOM | OOM |
| Gated LA Yang et al. (2023) | $f(x) = bx$ | $O(ND^2)$ | $O(ND)$ / $O(ND)$ | 100 | 5 |
| Our LA | $f(x) = a + bx$ | $O(ND^2)$ | $O(ND)$ / $O(ND)$ | 25 | 1.5 |

Table 1: Comparing the time and memory complexity of attention mechanisms. We achieve $3.3\times$ speedup and $3.6\times$ memory reduction compared to the sota linear attention (LA) implementation Gate LA. The forward-pass time and memory is for a single attention layer with a batch size of 4, 16 heads, dimension per head of 128, token length of $10^4$, and causal mask applied. (OOM = out of memory)

calculates its Output ($\mathbf{O}$) as a weighted sum of the Value vectors, where the weights are determined by the similarity between the Query of one position and the Key of all positions. This attention mechanism allows the model to weigh the importance of different parts of the input sequence when processing each position. The Output is then processed by a feed-forward network, which applies non-linear transformations. These layers are typically stacked multiple times, along with residual connections to build deep and powerful models. Our focus is solely on the attention mechanism.

**Notation:** Bold upper case letters, e.g., $\mathbf{X}$ indicate matrices, and bold lower case letters $\mathbf{x}_i$ and $\mathbf{x}_{ij}$ the $i$-th row and the element at the $i$-th row and $j$-th column of $\mathbf{X}$ respectively. Letters $X, x, x_i, x_{jkl}^{(i)}$ indicate scalars.

### 2.1 Regular Attention

In the Regular Attention mechanism, given a sequence of $N$ tokens, model dimension of $C$, $H$ heads, and dimension per head of $D = C/H$, each head takes in three matrices $\mathbf{Q}, \mathbf{K}, \mathbf{V} \in \mathbb{R}^{N \times D}$, and gives an output matrix $\mathbf{O} \in \mathbb{R}^{N \times D}$. The output is calculated using a matrix vector product (MVP) of the Attention matrix $\mathbf{A}$ with $\mathbf{V}$ as follows:

$$\mathbf{O} = \mathbf{A}\mathbf{V}, \ \mathbf{A} = \text{Softmax}\left(\mathbf{Q}\mathbf{K}^T\right), \tag{1}$$

$$\mathbf{o_{ij}} = \frac{\sum_{n=1}^{N} \exp(\mathbf{q}_i.\mathbf{k_n}/\sqrt{D})\mathbf{v_{n,j}}}{\sum_{n=1}^{N} \exp(\mathbf{q}_i.\mathbf{k}_n/\sqrt{D})} = \frac{\sum_{n=1}^{N} f(\mathbf{q}_i.\mathbf{k_n})\mathbf{v_{n,j}}}{\sum_{n=1}^{N} f(\mathbf{q}_i.\mathbf{k}_n)} \tag{2}$$

where $f(x)$ is the Attention kernel. Therefore, each head has a computational complexity of $O(N^2 D)$. The final output is given by concatenating outputs from each head. We can apply a causal mask by changing Eq. 2 to

$$\mathbf{O} = \text{tril}(\mathbf{A})\mathbf{V}, \quad \text{tril}(\mathbf{A})_{ij} = \begin{cases} \mathbf{a}_{ij}, j \leq i \\ 0, j > i \end{cases}, \quad \mathbf{o}_{ij} = \frac{\sum_{n=1}^{i} f(\mathbf{q}_i.\mathbf{k}_n)\mathbf{v}_{n,j}}{\sum_{n=1}^{i} f(\mathbf{q}_i.\mathbf{k}_n)}. \tag{3}$$

The traditional softmax attention kernel, defined by $f(x) = \exp(x/\sqrt{D})$, is not the sole viable option for deriving the attention matrix. In fact, any definite positive convex function can substitute for the attention kernel, with the choice impacting the model's expressivity.

### 2.2 Linear Attention

Using LA, the attention kernel is chosen to be $f(x) = a + bx$, with the coefficients either as the Taylor expansion of the exponential or as learnable parameters. The output with (left) and without (right) causal mask will be

$$\mathbf{o_{ij}} = \frac{\sum_{n=1}^{N}(a + b\mathbf{q}_i.\mathbf{k}_n)\mathbf{v_{n,j}}}{\sum_{n=1}^{N} a + b\mathbf{q}_i.\mathbf{k}_n}, \quad \mathbf{o_{ij}} = \frac{\sum_{n=1}^{i}(a + b\mathbf{q}_i.\mathbf{k}_n)\mathbf{v_{n,j}}}{\sum_{n=1}^{i} a + b\mathbf{q}_i.\mathbf{k}_n}, \tag{4}$$

which can be implemented using either an RNN architecture Katharopoulos et al. (2020); Yang et al. (2023) or a Transformer Han et al. (2023); Cai et al. (2022); Zhang et al. (2024); Wang et al.; Shen et al. (2021); You et al. (2024) in $O(ND^2)$ time. Our approach is based on the Transformer architecture. In Section 3 we demonstrate our novel method for calculating the forward and backward-pass of LA, and in Section 4 our optimized implementation.

## 3 Method

We analyze LA to identify and reuse repeated computation patterns, thereby reducing the computational cost of both the forward and backward-passes. We will demonstrate this process with a causal mask applied; attention without a causal mask follows a similar procedure. Our method and its subsequent implementation are intrinsically linked. The computational pattern we devise is specifically designed to enable a highly parallelized implementation with minimal data movement, as we elaborate in Section 4.

### 3.1 Forward-Pass

With $f(x) = a + bx$ as the attention kernel and causal mask applied, the output matrix $\mathbf{O}$ can be broken down to

$$\mathbf{o}_{ij} = \frac{\sum_{n=1}^{i} f(\mathbf{q}_i^T \mathbf{k_n}) \mathbf{v}_{nj}}{\sum_{n=1}^{i} f(\mathbf{q}_i^T \mathbf{k_n})} = \frac{\mathbf{f}_{ij}}{\mathbf{g}_i}, \tag{5}$$

$$\mathbf{f}_{ij} = \sum_{n=1}^{i} \left( a + b \sum_{m=1}^{D} \mathbf{q}_{im} \mathbf{k}_{n,m} \right) \mathbf{v}_{nj}, \quad \mathbf{g}_i = \sum_{n=1}^{i} \left( a + b \sum_{m=1}^{D} \mathbf{q}_{im} \mathbf{k}_{n,m} \right). \tag{6}$$

Changing the summation orders we get

$$\mathbf{f}_{ij} = a \sum_{n=1}^{i} \mathbf{v}_{nj} + b \sum_{m=1}^{D} \sum_{n=1}^{i} \mathbf{q}_{im} \mathbf{k}_{n,m} \mathbf{v}_{nj}, \qquad \mathbf{g}_i = a \sum_{n=1}^{i} 1 + b \sum_{m=1}^{D} \sum_{n=1}^{i} \mathbf{q}_{im} \mathbf{k}_{n,m}. \tag{7}$$

Factorizing the repeated computation we arrive at

$$\mathbf{f}_{ij} = x_{ij}^{(1)} + \sum_{m=1}^{D} \mathbf{q}_{im} x_{ijm}^{(2)}, \quad \mathbf{g}_i = y_i^{(1)} + \sum_{m=1}^{D} \mathbf{q}_{im} y_{im}^{(2)}, \tag{8}$$

where the $x, y$, are the identified repeated computation patterns and can be derived in linear time as

$$x_{1j}^{(1)} = a\,\mathbf{v}_{1j}, \quad x_{ij}^{(1)} = x_{i-1j}^{(1)} + a\,\mathbf{v}_{ij},$$

$$x_{1jm}^{(2)} = b\,\mathbf{k}_{1m}\mathbf{v}_{1j}, \quad x_{ijm}^{(2)} = x_{i-1jm}^{(2)} + b\,\mathbf{k}_{im}\mathbf{v}_{ij},$$

$$y_i^{(1)} = a\,i, \quad y_{1m}^{(2)} = b\,\mathbf{k}_{1m}, \quad y_{im}^{(2)} = y_{i-1m}^{(2)} + b\,\mathbf{k}_{im}. \tag{9}$$

Equation 5 requires $O(ND)$ operations, while Equations 7 and 9 both require $O(ND^2)$ operations. Therefore, the overall computational complexity of the forward-pass will be $O(ND^2)$. When using a differentiable programming library such as JAX or PyTorch, all operations must be tracked in the computational graph. Consequently, all intermediate variables from Eq. 9 need to be stored, resulting in a high memory consumption of $O(ND^2)$. To reduce the memory consumption and to also improve runtime efficiency, we derive the analytical gradient of the LA attention head and manually implement the backward-pass, instead of relying on these libraries.

### 3.2 Backward-Pass

To maintain an overall linear time scaling during training, the backward-pass must also be calculated with a time complexity of $O(ND^2)$. Let us denote the dot product $\mathbf{q}_i.\mathbf{k}_j$ as $s_{ij}$, and write the attention $\mathbf{a}$ and output $\mathbf{o}$ with causal mask applied as

$$\mathbf{o}_{ij} = \sum_{n=1}^{i} \mathbf{a}_{in} \mathbf{v}_{nj}, \quad \mathbf{a}_{in} = \frac{f(s_{ij})}{\sum_{m=1}^{i} f(s_{im})} = \frac{f(s_{ij})}{\mathbf{g}_i}, \quad f(x) = a + bx \tag{10}$$

Taking the derivative with respect to $s_{il}$, we find $\dfrac{\partial \mathbf{o}_{ij}}{\partial s_{il}}$ as

$$\frac{\partial \mathbf{a}_{in}}{\partial s_{il}} = \begin{cases} \dfrac{b\,(1 - \mathbf{a}_{ij})}{\sum_{m=1}^{i} f(s_{im})}, & n = l \\[3mm] \dfrac{b\,(-\mathbf{a}_{ij})}{\sum_{m=1}^{i} f(s_{im})}, & n \neq l \end{cases} \tag{11}$$

$$\frac{\partial \mathbf{o}_{ij}}{\partial s_{il}} = \sum_{n=1}^{i} \frac{\partial \mathbf{a}_{in}}{\partial s_{il}} \mathbf{v}_{nj} = \frac{b\,\left( \mathbf{v}_{lj} - \sum_{n=1}^{i} \mathbf{a}_{in} \mathbf{v}_{nj} \right)}{\sum_{m=1}^{i} f(s_{im})} = \frac{b}{\mathbf{g}_i} (\mathbf{v}_{lj} - \mathbf{o}_{ij}). \tag{12}$$

We now derive the partial derivative with respect to $\mathbf{Q}, \mathbf{K}, \mathbf{V}$

$$\frac{\partial \mathbf{o}_{ij}}{\partial \mathbf{q}_{ir}} = \sum_{l=1}^{i} \frac{\partial s_{il}}{\partial \mathbf{q}_{ir}} \frac{\partial \mathbf{o}_{ij}}{\partial s_{il}} = \frac{\sum_{l=1}^{i} b\,\mathbf{k}_{l,r}}{\mathbf{g}_i}(\mathbf{v}_{lj} - \mathbf{o}_{ij}) \tag{13}$$

$$\frac{\partial \mathbf{o}_{ij}}{\partial \mathbf{k}_{p,r}} = \frac{\partial s_{ip}}{\partial \mathbf{k}_{p,r}} \frac{\partial \mathbf{o}_{ij}}{\partial s_{ip}} = \frac{b\,\mathbf{q}_{ir}}{\mathbf{g}_i}(\mathbf{v}_{p,j} - \mathbf{o}_{ij}) \tag{14}$$

$$\frac{\partial \mathbf{o}_{ij}}{\partial \mathbf{v}_{p,j}} = \mathbf{a}_{ip} = \frac{f(s_{ip})}{\mathbf{g}_i}. \tag{15}$$

During the backward-pass, given the gradient of the layer in front $\mathbf{\Omega}$, the gradient of the attention head $\nabla\mathbf{\Psi}$ is calculated as follows

$$\nabla_{\mathbf{q}_{ir}}\mathbf{\Psi} = \sum_{j=1}^{D} \frac{\partial \mathbf{o}_{ij}}{\partial \mathbf{q}_{ir}}\mathbf{\Omega}_{ij} = \sum_{j=1}^{D} \frac{\sum_{l=1}^{i} b\,\mathbf{k}_{l,r}(\mathbf{v}_{lj} - \mathbf{o}_{ij})}{\mathbf{g}_i}\mathbf{\Omega}_{ij} \tag{16}$$

$$\nabla_{\mathbf{k}_{p,r}}\mathbf{\Psi} = \sum_{i=1}^{p}\sum_{j=1}^{D} \frac{\partial \mathbf{o}_{ij}}{\partial \mathbf{k}_{p,r}}\mathbf{\Omega}_{ij} = \sum_{i=1}^{p}\sum_{j=1}^{D} \frac{b\,\mathbf{q}_{ir}}{\mathbf{g}_i}(\mathbf{v}_{p,j} - \mathbf{o}_{ij})\,\mathbf{\Omega}_{ij} \tag{17}$$

$$\nabla_{\mathbf{v}_{p,j}}\mathbf{\Psi} = \sum_{i=1}^{p} \frac{\partial \mathbf{o}_{ij}}{\partial \mathbf{v}_{p,j}}\mathbf{\Omega}_{ij} = \sum_{i=1}^{p} \frac{f(s_{ip})}{\mathbf{g}_i}\,\mathbf{\Omega}_{ij} \tag{18}$$

Changing the summation order and factorizing the repeated computation we get

$$\nabla_{\mathbf{q}_{ir}}\mathbf{\Psi} = \sum_{j=1}^{D} \alpha_{ijr}^{Q} - \beta_{ir}^{Q}\,\mathbf{o}_{ij}\,\hat{\mathbf{\Omega}}_{ij}, \qquad \nabla_{\mathbf{k}_{ir}}\mathbf{\Psi} = \sum_{j=1}^{D} \alpha_{irj}^{K}\,\mathbf{v}_{ij} - \beta_{irj}^{K}, \tag{19}$$

$$\nabla_{\mathbf{v}_{ij}}\mathbf{\Psi} = \alpha_{ij}^{V} + \sum_{j=1}^{D} \beta_{rj}^{V}\,\mathbf{k}_{ir}, \qquad \hat{\mathbf{\Omega}}_{ij} = \frac{\mathbf{\Omega}_{ij}}{\mathbf{g}_i}. \tag{20}$$

where the $\alpha$ and $\beta$ are the factorized coefficients and are defined as

$$\begin{aligned}
\alpha_{1jr}^{Q} &= b\,\mathbf{k}_{1r}\,\mathbf{v}_{1j}, & \alpha_{ijr}^{Q} &= \alpha_{i-1jr}^{Q} + b\,\mathbf{k}_{ir}\,\mathbf{v}_{ij}, \\
\beta_{1r}^{Q} &= b\,\mathbf{k}_{1r}, & \beta_{ir}^{Q} &= \beta_{i-1r}^{Q} + b\,\mathbf{k}_{ir}, \\
\alpha_{Nrj}^{K} &= b\,\mathbf{q}_{Nr}\,\hat{\mathbf{\Omega}}_{Nj}, & \alpha_{irj}^{K} &= \alpha_{i+1rj}^{K} + b\,\mathbf{q}_{ir}\,\hat{\mathbf{\Omega}}_{ij}, \\
\beta_{Nrj}^{K} &= b\,\mathbf{q}_{Nr}\,\mathbf{o}_{Nj}\,\hat{\mathbf{\Omega}}_{Nj}, & \beta_{irj}^{K} &= \beta_{i+1rj}^{K} + b\,\mathbf{q}_{ir}\,\mathbf{o}_{ij}\,\hat{\mathbf{\Omega}}_{ij}, \\
\alpha_{Nj}^{V} &= a\,\hat{\mathbf{\Omega}}_{Nj}, & \alpha_{Nj}^{V} &= \alpha_{i+1j}^{V} + a\,\hat{\mathbf{\Omega}}_{ij}, \\
\beta_{Nrj}^{V} &= b,\ \mathbf{q}_{Nr}\,\hat{\mathbf{\Omega}}_{Nj}, & \beta_{irj}^{V} &= \beta_{i+1rj}^{V} + b,\ \mathbf{q}_{ir}\,\hat{\mathbf{\Omega}}_{ij}.
\end{aligned} \tag{21}$$

Put into words, gradient of the output matrix $\mathbf{O}$ can be calculated by storing $\mathbf{Q}$, $\mathbf{K}$, $\mathbf{V}$, $\mathbf{O}$, and $\mathbf{g}_i$; resulting in a reduced memory consumption of $O(ND)$ elements. The time complexity is $O(ND^2)$ since we perform $O(D)$ operations for $1 \le i \le N$, $1 \le j \le D$ in Equation 21, which is the same time complexity as the forward-pass.

### 3.3 Normalization

It has been suggested Qin et al. (2022a) that the lack of expressivity in other subquadratic attention mechanisms rises due to the possibility of their gradients either blowing up or vanishing. To prevent such behavior, we recommend normalizing the query and key vectors before the attention calculation. To be specific, given $\mathbf{Q}$ and $\mathbf{K}$, we first normalize $\mathbf{Q}$ and $\mathbf{K}$ row-wise as

$$\mathbf{q}_i = \frac{\mathbf{q}_i}{||\mathbf{q}_i||}, \quad \mathbf{k}_i = \frac{\mathbf{k}_i}{||\mathbf{k}_i||}, \quad \forall 1 \le i \le N. \tag{22}$$

## 4 Optimized Implementation

Calculating LA in the specific format detailed in Section 3 offers the potential to increase both the data reuse rate (the number of times each element is used per off-chip memory access) and parallelization. Specifically, the operations are ordered to ensure that each data segment is accessed by a single thread. This ordering prevents scenarios where multiple threads conflict over the same element or where individual threads require data exceeding the shared memory (L1 cache) size, thereby preventing redundant off-chip memory accesses. In this section we will provide the details of our optimized CUDA implementation such as data movement and thread scheduling for the forward and backward-pass. We employ attention kernel of $f(x) = 1 + x$, and the variables are stored as three dimensional tensors.

### 4.1 Forward-Pass

To calculate the output $\mathbf{O}$, we require $\mathbf{f}_{ij}$ and $\mathbf{g}_i$, where $\mathbf{f}_{ij}$ and $\mathbf{g}_i$ are obtained according to Equation 8. We will detail the process for calculating $\mathbf{f}_{ij}$, and $\mathbf{g}_i$ follows a similar, simpler process.

To derive $\mathbf{f}_{ij}$, we need to find two sets of coefficients $x^{(1)}$ and $x^{(2)}$, as described by Equation 9. The derivation of $x^{(1)}$ and $x^{(2)}$ is sequential, meaning that $x_{ij}^{(1)}$ and $x_{ijm}^{(2)}$ depend on $x_{i-1j}^{(1)}$ and $x_{i-1j}^{(2)}$. As a result, the derivation of each individual $x_{ij}^{(1)}$ and $x_{ijm}^{(2)}$ cannot be parallelized; at least not without employing '*reduction*' techniques[2]. For now, let's assume a non-parallelized approach. Similarly, the derivation of each individual $f_{ij}$ is also inherently sequential. Therefore, the computation of $x_{ij}^{(1)}$, $x_{ijm}^{(2)}$ and $f_{ij}$ for each $j$ should be performed within the same thread as well.

As a reminder $\mathbf{f}_{ij}$ is calculated as

$$\mathbf{f}_{ij} = x_j^{(1)} + \sum_{m=1}^{D} \mathbf{q}_{im} x_{jm}^{(2)}, \tag{23}$$

which we refer to the left term $x_j^{(1)}$ as the Constant term and the right term $\sum_{m=1}^{D} \mathbf{q}_{i,m} x_{jm}^{(2)}$ as the Linear term. Let's first proceed with the Constant term. We call $D$ threads for $1 \le j \le D$, in each thread perform a loop for $1 \le i \le N$ to calculate $x_{ij}^{(1)}$, and assign it to $f_{ij}$ as shown in Algorithm 1 in the Appendix. Since the $x_{ij}^{(1)}$ values for each $j$ is only accessed within a thread, we can use a single register to store and update the $x_{ij}^{(1)}$ values in the for loop, rather than store them for each $i$. This will accelerate the read operations (accessing in-thread registers is considerably faster than accessing shared memory) and reduce the memory consumption. Furthermore, each thread $j$ will access the off-chip memory to read the $\mathbf{v}_{ij}$ values for $1 \le i \le N$, and to write $\mathbf{f}_{ij}$. To leverage the efficient data movement capabilities of NVIDIA GPU Streaming Multiprocessors (SMs), which fetch adjacent data blocks from off-chip memory to the L2 and L1 caches for adjacent blocks and threads, the $\mathbf{V}_{ij}$ and $\mathbf{f}_{ij}$ values should be stored with $j$ as the first tensor dimension and $i$ as the second.

To calculate the Linear term, we call $D$ threads for $1 \le j \le D$, and in each thread perform a loop for $1 \le m \le D$ within a loop for $1 \le i \le N$ to calculate $x_{ijm}^{(2)}$ and add it to $f_{ij}$, as shown in Algorithm 2. Since the $x_{ijm}^{(2)}$ values for each $j$ are only accessed within a thread, we can use an array of registers to store them for each $m$, and sequentially update it as the $x_{ijm}^{(2)}$ values to accelerate read operations. However, the value of $D$ is often too large for the whole $x_{ijm}^{(2)}$ $1 \le m \le D$ to be stored in thread registers. As a workaround we perform reduction, where we break the inner loop into separate $L$ block of threads, that is, we call $L$ blocks, each with $D$ threads, and in each thread performing loop for $1 \le m \le D/L$ within a loop for $1 \le i \le N$. By applying reduction, multiple threads might edit the same $f_{ij}$, and therefore we would have to use 'Race Aware' operations when updating $f_{ij}$. We use $L = \dfrac{D}{32}$ in our implementation.

Furthermore, since the $\mathbf{q}_{im}$ and $\mathbf{k}_{im}$ values for each $m$ are accessed within the same block of threads, we can store them in shared memory (note that multiple block of threads will access the same $\mathbf{q}_{im}$ and $\mathbf{k}_{im}$). To be

---

[2]https://developer.download.nvidia.com/assets/cuda/files/reduction.pdf

specific, within every iteration of the outer loop (for $1 \leq i \leq n$), each thread moves one element of $\mathbf{q}$ and $\mathbf{k}$ from the off-chip memory to the shared memory; a total of $2 \times D$ elements. Since $64 \leq D \leq 512$, there is enough shared memory for storage. In order to enable the adjacent threads utilize the shared memory, corresponding $\mathbf{q}_{im}$ and $\mathbf{k}_{im}$ values should be stored with $m$ as the first tensor dimension and $i$ as the second. By parallelizing the read operations and utilizing shared memory, we reduce the read time from $ND\,t_{oc}$ to $N, t_{oc} + D\,t_s$, where $t_{oc}$ and $t_s$ are the read time for off-chip and shared memory ($t_s << t_{oc}$). For context, $t_{oc}$ is on the order of a hundred cycles, where $t_s$ a few cycles.

Finally, considering the multiple attention heads $H$ in each Transformer layer and multiple input batches $B$, the entire process is executed across $B \times H$ parallel blocks. The values are stored such that the third tensor dimension corresponds to the batch and head numbers. Overall, the process involves $B \times H$ outer blocks, $L$ inner blocks, and $D$ threads per block. For instance, Figure 1 illustrates the structure of the $\mathbf{q}_{rij}$ values and the data movement pattern for calculating the Linear term with a batch size of $B = 2$, number of heads $H = 4$, dimension per head $D = 6$, sequence length $N = 8$, and *reduction* block size $L = 2$ (indicated as A and B blocks), where $1 \leq r \leq B \times H$, $1 \leq i \leq N$, and $1 \leq j \leq D$. Cells sharing the same color belong to the same outer block, cells with the same letter (A or B) belong to the same inner block, and each cell within an inner block is accessed by D threads. The $\mathbf{q}$ cells are then processed sequentially, with the left number in each cell indicating the outer loop iteration (line 11 in Algorithm 2) and the right number indicating the inner loop iteration (line 16 in Algorithm 2). In a 1-dimensional view, the data are arranged so that data processed by adjacent threads are spatially adjacent. For example, cells with the same color and the same letter are positioned next to each other.

Note that the data movement differs for other tensors such as $\mathbf{k}$ and $\mathbf{v}$. We should also mention that we must ensure threads related to the Constant term complete their execution before invoking threads related to the Linear term to prevent race conditions, as both sets of threads might modify the same $f_{ij}$.

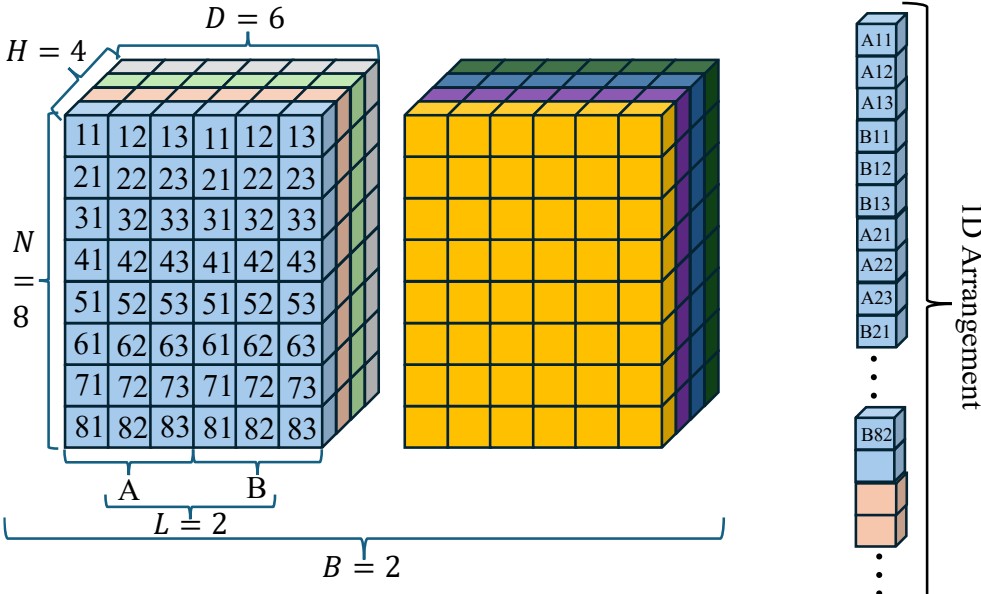

Figure 1: Structure of the Query ($\mathbf{Q}$) storage and its data movement pattern for calculating the Linear term with a batch size of $B = 2$, number of heads $H = 4$, dimension per head $D = 6$, sequence length $N = 8$, and *reduction* block size $L = 2$. Cells with the same color and letter (A or B) are processed by the same block of threads, and the left and right numbers on each cell denote the sequential order of processing in each thread's outer and inner loops.

### 4.2 Backward-Pass

During the backward-pass, the attention layer takes the incoming gradient of the layer in front, and produces the gradient with respect to $\mathbf{Q}$, $\mathbf{K}$ and $\mathbf{V}$, as shown in Equations 19-20. We will detail the process for calculating the gradient with respect to $\mathbf{K}$ ($\nabla_{\mathbf{K}}\boldsymbol{\Psi}$), and $\nabla_{\mathbf{Q}}\boldsymbol{\Psi}$, $\nabla_{\mathbf{V}}\boldsymbol{\Psi}$ follows a similar process. To derive $\mathbf{f}_{ij}$, we need to find two sets of coefficients $\alpha^K$ and $\beta^K$. Similar to the forward-pass, these coefficients are determined sequentially, as described by Equation 21. As a reminder $\nabla_{\mathbf{k}_{ir}}\boldsymbol{\Psi}$ is calculated as

$$\nabla_{\mathbf{k}_{ij}}\boldsymbol{\Psi} = \sum_{m=1}^{D} \alpha_{ijm}^K \mathbf{v}_{im} - \sum_{m=1}^{D} \beta_{ijm}^K, \tag{24}$$

where we refer to the left term $\sum_{m=1}^{D} \alpha_{ijm}^K \mathbf{v}_{im}$ as the Alpha term, and the right term $\sum_{m=1}^{D} \beta_{ijm}^K$ the Beta term. To derive them, we take an approach similar to calculation of Linear term in forward-pass; to call $B \times H$ outer blocks, $L$ inner blocks, and $D$ threads per block.

To calculate the Alpha term, each thread consists of an outer loop for $1 \le i \le N$ and an inner loop for $1 \le m \le D/L$. The $\alpha_{ijm}^K$ are then calculated sequentially and assigned to $\nabla_{\mathbf{k}_{ij}}\boldsymbol{\Psi}$, as shown in Algorithm 3. Given that the $\mathbf{q}_{ij}$ values for each $j$ are solely accessed within a thread, we can use a single register to store and update them in the for loop. Furthermore, since the $\alpha_{ijm}^K$ values are accessed only within a thread as well, we use an array of registers to store them, employing the same *reduction* techniques to manage the limited register space.

The $\mathbf{v}_{im}$ and $\hat{\boldsymbol{\Omega}}_{im}$ for each $m$ are accessed within the same block of threads. Therefore, we can store them in shared memory. Analogous to the Linear terms in forward pass, the data movement from the off-chip to shared memory is parallelized among each block of threads to reduce the read time from $ND\,t_{oc}$ to $N, t_{oc} + D\,t_s$. To optimize data movement for $\hat{\boldsymbol{\Omega}}_{ij}$ and $\nabla_{\mathbf{k}_{ij}}\boldsymbol{\Psi}$ by the SMs, we should store them with $m$ and $i$ mapping to the first and second tensor dimensions respectively. This arrangement allows adjacent threads to access adjacent memory, enhancing efficiency.

Similarly, to calculate the Beta term, each thread consists of an outer loop for $1 \le i \le N$ and an inner loop for $1 \le m \le D/L$, where the $\beta{ijm}^K$ are calculated sequentially and deducted from $\nabla_{\mathbf{k}_{ij}}\boldsymbol{\Psi}$, as shown in Algorithm 4. The main difference compared to Alpha term calculation is that instead of storing $\mathbf{v}_{im}$ and $\hat{\boldsymbol{\Omega}}_{im}$ in the shared memory, $\mathbf{o}_{im}$ and $\hat{\boldsymbol{\Omega}}_{im}$ will be stored.

## 5 Evaluation

We evaluate our work through two sets of evaluations: first, in an isolated setting as a standalone attention layer with causal mask applied, and second, in an end-to-end setting where we train an LLM. We refer to our implementation as 'Our LA', and compare our results with Gated LA Yang et al. (2023) (the state-of-the-art LA implementation), Speculative Decoding (Spec. Dec.) LA You et al. (2024), and baseline LA where we use the default Pytorch operations. We also compare with efficient I/O aware implementation of Regular Attention, FlashAttention-2 Dao (2024).

### 5.1 Standalone Layer

The upper plots in Figure 2 show a log-log analysis of the time and memory complexity of a single attention head's forward-pass as a function of sequence length $N$, ranging from $10^3$ to $3\ddot{O}10^5$ tokens. These evaluations were conducted with a batch size of $B = 4$, number of heads $H = 16$ attention heads, and a per-head dimension of $D = 128$, with the reported results representing the average of 100 iterations performed on an NVIDIA A6000 GPU and precision of float-32. The slopes observed in the plots indicate the order of dependency on $N$. All LA implementations exhibit a runtime scaling linearly with the number of tokens, while Regular Attention scales quadratically. Given the current trend of transformers scaling to $\sim 10^7$ context length, LA would be substantially more efficient. Furthermore, our implementation is 3.3× faster than Gated LA. The observed performance improvement is attributed to our reduced data movement, and the implementation of LA using the Transformer architecture, in contrast to Gated LA's use of an RNN.

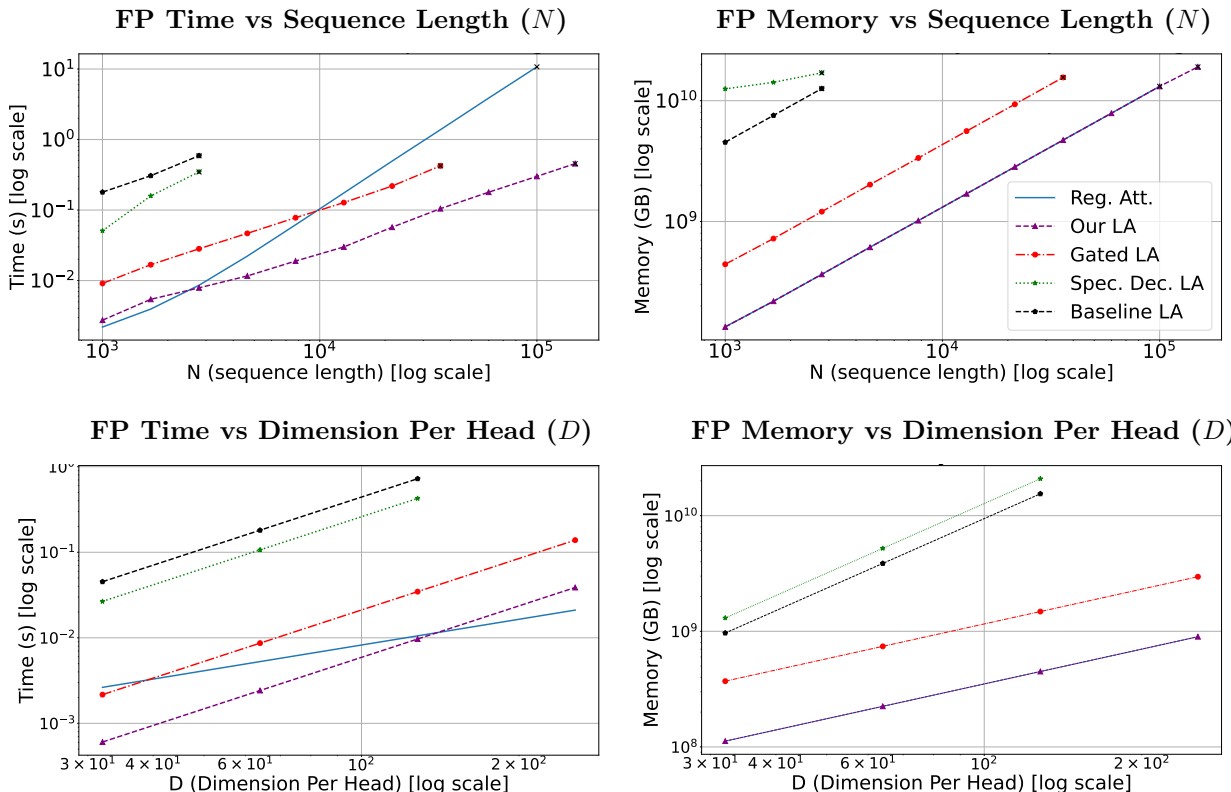

Figure 2: Time and memory scaling of our LA implementation, FlashAttention-2 Dao (2024) (Regular Attention), Gated LA Yang et al. (2023), Speculative Decoding LA You et al. (2024), and baseline Pytorch LA implementation on an A6000 GPU during forward-pass (FP). The top two figures show the time and memory scaling as a function of number of tokens $N$ for dimension per head of $D = 128$, batch size $B = 4$, and number of heads $H = 16$. The bottom two the figures show the scaling as a function of $D$ for $N = 4096$. The 'Reg. Att.' and 'Our LA' lines overlap in the memory scaling plots.

Specifically, the inherent sequential structure of RNNs results in sequential computation of the output matrix, thereby limiting parallelization. In contrast, Transformers inherently allow for parallel computation of the output matrix. Moreover, the baseline and Spec. Dec. LA, which are based on Transformer architecture, are an order of magnitude slower, showcasing the effectiveness of our method and implementation. Regarding memory consumption, all implementations scale linearly with $N$. Our LA and Regular Attention have the lowest consumption, achieving $3.6\times$ less memory consumption than Gated LA and an order of magnitude lower than the baseline and Spec. Dec. LA.

The bottom plots in Figure 2 show the time and memory scaling of an attention head against dimension per head $D$ under the same setting, except $32 \leq D \leq 256$ and $N = 4096$. All of the LA implementations have a runtime scaling quadratcially with dimension per head, whereas Regular Attention scales linearly. Fortunately, the dimension per head is a constant parameter ($10^2$) in the Transformer architecture, and is relatively small ($D << N$). As for the memory consumption, Our LA, Gated LA and Regular Attention have a linear scaling with $D$, whereas baseline and Spec. Dec. LA scale quadratically. This quadratic scaling arises from the application of a causal mask. To be specific, relying on a differentiable programming library to derive the backward-pass for LA with a causal mask results in $O(ND^2)$ memory complexity, as discussed in Section 3.2. In contrast, Our LA and Gated LA employ manually derived backward-passes.

The upper and bottom plots in Figure 3 show the time and memory scaling for a single backward-pass under the same setting as above against $N$ and $D$ respectively. The scaling trends mirror those observed in the forward-pass. Specifically, Regular Attention exhibits a quadratic runtime scaling, while the LA

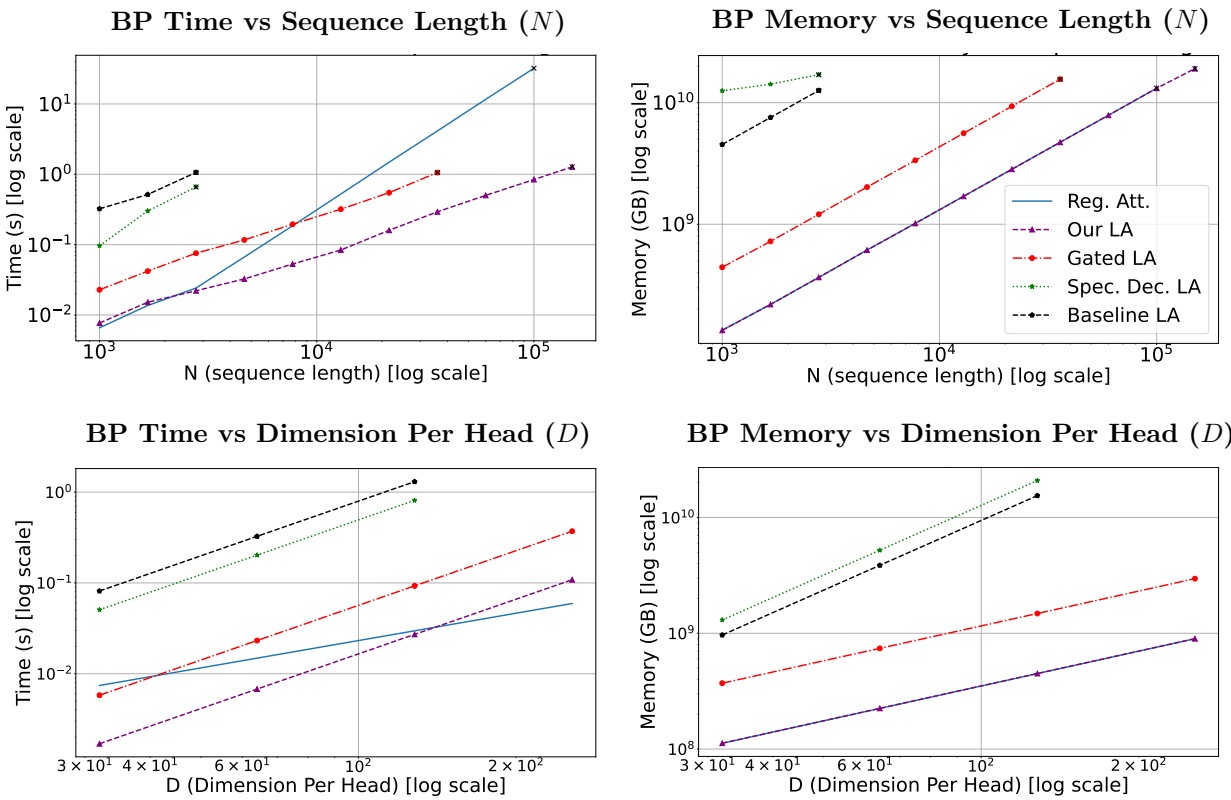

Figure 3: Time and memory scaling of our LA implementation, FlashAttention-2 Dao (2024) (Regular Attention), Gated LA Yang et al. (2023), Speculative Decoding LA You et al. (2024), and baseline Pytorch LA implementation on an A6000 GPU during backward-pass (BP). The upper two figures show the time and memory scaling as a function of number of tokens $N$ for dimension per head of $D = 128$, batch size $B = 4$, and number of heads $H = 16$. The bottom two the figures show the scaling as a function of $D$ for $N = 4096$. The 'Reg. Att.' and 'Our LA' lines overlap in the memory scaling plots.

implementations demonstrate linear scaling, with our implementation significantly outperforming other LA variants in terms of time and memory consumption.

Figure 4 presents the ratio of time spent on data movement (primarily off-chip memory read/write) to the total runtime (left), and the total data movement time (right) for LA implementations averaged over 100 runs on an A6000. Our method allows for a computational pattern that maximizes data reuse rate, and our implementation achieves this efficiency through specific thread scheduling and data movement, as elaborated in Section 4. As a result, our implementation exhibits minimal data movement. Compared to other LA implementations, the portion of time spent on data movement is significantly lower, where it is roughly one-third of the next lowest ratio, observed in Gated LA (71%). This optimized data movement becomes even more apparent when examining the total data movement time, where our implementation achieves an order of magnitude reduction compared to the next best, Gated LA.

Compared to baseline LA implementation, where default Pytorch functions are used, the difference is a staggering 100×. This significant gap arises from the inherent limitations of differentiable programming libraries. More precisely, to derive gradients during the backward-pass, these libraries must store every operation within a computational graph and apply the 'Chain Rule'. This approach becomes highly inefficient for element-wise operations, especially with extremely large data dimensions, as storing operations for each element results in a massive graph. Instead, tensor-wise operations should be employed, where the same operation is applied to the entire tensor and can be stored efficiently in the graph. However, the drawback of tensor-wise operations is that they restrict the flexibility of calculation patterns, and each such operation

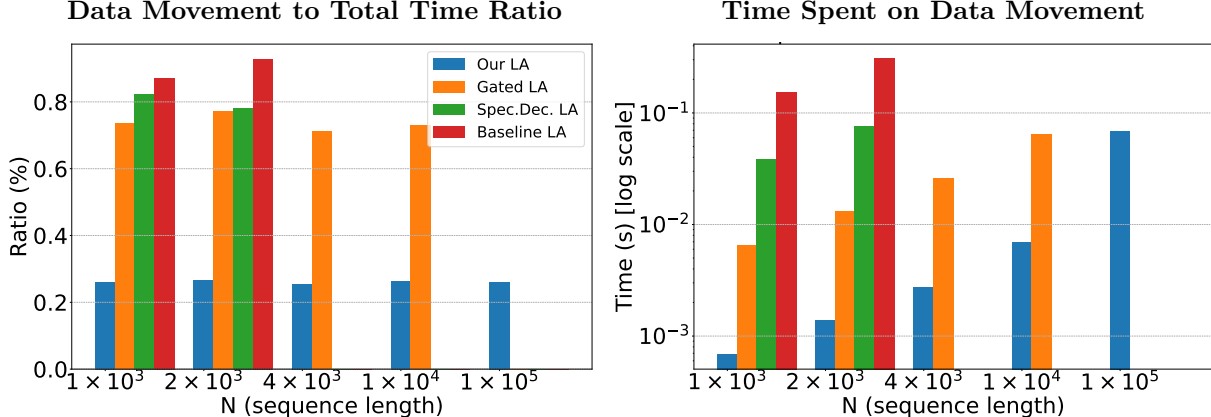

Figure 4: Ratio of data movement time to total runtime (left), and total data movement time for the forward-pass (right) across various sequence lengths for LA implementations. Empty bars indicate out of memory. Our approach minimizes data movement, resulting in significantly lower ratio and movement time.

necessitates a read/write access to off-chip memory, which is extremely slow and can take hundreds of cycles. On the A6000 architecture, off-chip memory access typically ranges from 300 to 800 cycles, depending on memory traffic. In contrast, a custom CUDA implementation grants us the freedom to implement intricate computational and data movement patterns, which in turn enables optimized implementations, such as the one described in Section 4.

In conclusion, our implementation scales linearly with $N$ and $D$ in terms of both time and memory, achieving a $3.3\times$ faster runtime and $3.6\times$ lower memory consumption compared to Gated LA, the state-of-the-art LA implementation. In comparison to efficient I/O-aware Regular Attention implementation FlashAttention-2, our approach demonstrates a faster runtime for $N > 3000$ while maintaining similar memory consumption.

## 5.2 Training an LLM

To demonstrate the usefulness of our LA in practice, we train an LLM on the English partition of Wiki-40B Guo et al. (2020). We chose Pythia-1.4B Biderman et al. (2023) as our LLM architecture with a token length of $N = 8192$. We deployed our model using LitGPT AI (2023) using eight A6000 GPUs. We use rotary positional encoding Su et al. (2024), cosine warmup and decay with a minimum and maximum learning rate of $5e - 5$ and $1e - 3$, and global batch size of 256. We train the model using our implementation of LA, Gated LA, and Regular Attention (FlashAttention-2). The hyperparameters used are the same for both methods. The plots in Figure 5 show the learning curves for training loss as a function of wall-clock time (left) and training step (right). Compared to Regular Attention, both LA implementations have a slower convergence based on iteration count. However, they converge to the same loss, hinting to the fact that LA has comparable expresivity to Regular Attention. As for wall-clock time, our LA implementation exhibits substantial speedup of $2.8\times$ compared to Gated LA. Compared to Regular Attention, we exhibit $1.8\times$ speedup. The latter is attributed to $O(N)$ time scaling of LA, as opposed to Regular Attention's scaling of $O(N^2)$.

Table 2 shows the comparison between LA implementations and Regular Attention in terms of accuracy on a the MMLU Hendrycks et al. (2020), PIQA Bisk et al. (2020), and ARC Clark et al. (2018) benchmarks. In agreement to results demonstrated by previous work Yang et al. (2023); You et al. (2024); Katharopoulos et al. (2020); Han et al. (2023); Cai et al. (2022); Zhang et al. (2024); Wang et al.; Shen et al. (2021); Arora et al. (2024), LA is able to achieve comparable results to Regular Attention. This is significant as it implies that LA holds promise for replacing Regular Attention in resource-constrained devices such as smartphones, laptops, and edge devices. Our implementation of LA offers a substantial speedup over both state-of-the art LA implementation and efficient I/O aware Regular Attention in an end to end LLM setting, facilitating practical usage of LA.

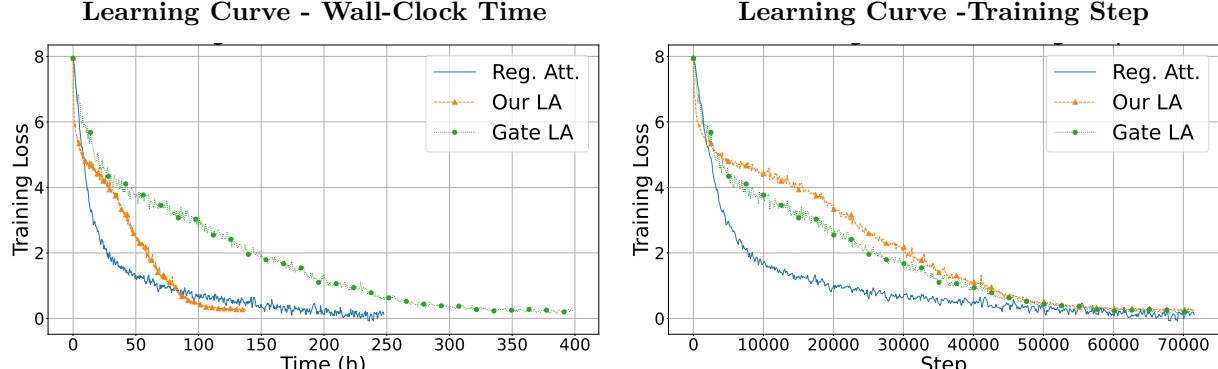

Figure 5: The learning curves of Pythia-1.4B, trained on the English segment of the Wiki-40B with a context length of 8192 using eight A6000 GPUs. The learning curves are presented as a function of wall-clock time (left) and training step (right). Both models were trained for the same number of steps. Our linear attention (LA) implementation offers speedup over Regular Attention and Gated LA. The loss measures cross-entropy.

Table 2: Comparing accuracy of Regular Attention and linear attention (LA) implementations. Unlike regular attention, which scales at $O(N^2)$ with the number of tokens $N$, LA scales linearly at $O(N)$ while achieving comparable accuracy.

| Mechanism | MMLU (0-shot) | MMLU (5-shot) | PIQA | ARC-C | ARC-E | ARC-E |
|---|---|---|---|---|---|---|
| Regular Attention | 22.10 | 23.04 | 69.18 | 33.83 | 61.12 | 60.34 |
| Gated LA Yang et al. (2023) | 21.15 | 22.27 | 65.38 | 30.52 | 57.73 | 58.12 |
| **Our LA** | 21.16 | 25.90 | 68.63 | 32.43 | 59.95 | 59.08 |

## 6 Conclusion

Transformers use a Softmax-based mechanism known as attention, which has a *quadratic* scaling with number of tokens taken in context. This problem is particularly important since the current applications are moving towards higher number of tokens, and that there is significant interest in deploying large language models on devices with limited computational power such as smart phones. Consequently, the search for an efficient attention mechanism has become an important and active area of research. One of the promising approaches is Linear Attention (LA) with a linear time complexity of $O(ND^2)$, where $N$ is the number of tokens and $D$ the dimension per attention head. However, due to the less established nature of LA, its implementation is far less optimized compared to regular attention and there is much room for improvement. We present an extensively optimized LA implementation.

In summary, we propose a novel approach for calculating LA, use it to derive the analytical gradient of Attention heads, demonstrate how forward and backward passes can be calculated in $O(N)$ time, and provide the details of our optimized implementation. Our method and its implementation work hand-in-hand in the sense that the computational pattern we devise allows for a highly parallelized implementation with minimal data movement. We evaluate our implementation in an isolated setting as a standalone attention layer, and in an end-to-end setting where we train an LLM. We achieve 3.3× speedup and a 3.6× reduction in memory consumption compared to the state-of-the-art LA implementation Yang et al. (2023). Our implementation is in format of a PyTorch library, and can be used as a plug-and-play module. By improving the efficiency of LA implementation, we are paving the path for its practical usage, which will be particularly beneficial for deploying LLMs on computationally constrained devices.

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

# A  Implementation Pseudo Codes

This appendix provides the CUDA pseudo codes of our implementations referenced in Section 4

---

**Algorithm 1** Forward-Pass, Constant term

---

1: **Input:** v $\in \mathbb{R}^{B \times H, N, D}$
2: **Output:** f $\in \mathbb{R}^{B \times H, N, D}$
3: **Call** $B \times H$ blocks ($1 \leq bh \leq B \times H$)
4: **Call** $D$ threads in each block ($1 \leq j \leq D$)
5: In each thread:
6: x = 0 (on-thread register)
7: **for** $i \leftarrow 1$ **to** $N$ **do**
8:    x += v[bh][i][j]
9:    f[bh][i][j] = x
10: **end for**

---

**Algorithm 2** Forward-Pass, Linear term

---

1: **Input:** q, k, v $\in \mathbb{R}^{B \times H, N, D}$
2: **Output:** f $\in \mathbb{R}^{B \times H, N, D}$
3: **Call** $B \times H$ outer blocks ($1 \leq bh \leq B \times H$)
4: **Call** $L$ inner blocks ($1 \leq l \leq L$)
5: **Call** $D$ threads in each block ($1 \leq j \leq D$)
6: In each thread:
7: vr, x = 0 (on-thread register)
8: r[1 to $D/L$] = 0 (on-thread register)
9: sq[1 to $D$] = 0 (shared memory)
10: sk[1 to $D$] = 0 (shared memory)
11: **for** $i \leftarrow 1$ **to** $N$ **do**
12:    x = 0
13:    vr = v[bh][i][j]
14:    **load** k[bh][i][j] **to** sk[j]
15:    **load** q[bh][i][j] **to** sq[j]
16:    **for** $m \leftarrow 1$ **to** $D/L$ **do**
17:      r[m] += vr×sk[$l \times D/L + m$]
18:      x += r[m]×sq[$l \times D/L + m$]
19:    **end for**
20:    f[bh][i][j] += x
21: **end for**

---

---

**Algorithm 3** Backward-Pass, Alpha term

---

1: **Input:** q, v, $\hat{\Omega} \in \mathbb{R}^{B \times H, N, D}$
2: **Output:** $\nabla_{\mathbf{k}}\boldsymbol{\Psi} \in \mathbb{R}^{B \times H, N, D}$
3: **Call** $B \times H$ outer blocks ($1 \leq bh \leq B \times H$)
4: **Call** $L$ inner blocks ($1 \leq l \leq L$)
5: **Call** $D$ threads in each block ($1 \leq j \leq D$)
6: In each thread:
7: qr, x = 0 (on-thread register)
8: r[1 to $D/L$] = 0 (on-thread register)
9: sv[1 to $D$] = 0 (shared memory)
10: sg[1 to $D$] = 0 (shared memory)
11: **for** $i \leftarrow N$ **to** 1 **do**
12:     x = 0
13:     qr = q[$bh$][$i$][$j$]
14:     **load** v[$bh$][$i$][$j$] **to** sv[$j$]
15:     **load** $\hat{\Omega}$[$bh$][$i$][$j$] **to** sg[$j$]
16:     **for** $m \leftarrow 1$ **to** $D/L$ **do**
17:         r[$m$] += qr×sg[$l \times D/L + m$]
18:         x += r[$m$]×sv[$l \times D/L + m$]
19:     **end for**
20:     $\nabla_{\mathbf{k}}\boldsymbol{\Psi}$[bh][i][j] = x
21: **end for**

---

---

**Algorithm 4** Backward-Pass, Beta term

---

1: **Input:** q, v, $\hat{\Omega} \in \mathbb{R}^{B \times H, N, D}$
2: **Output:** $\nabla_{\mathbf{k}}\boldsymbol{\Psi} \in \mathbb{R}^{B \times H, N, D}$
3: **Call** $B \times H$ outer blocks ($1 \leq bh \leq B \times H$)
4: **Call** $L$ inner blocks ($1 \leq l \leq L$)
5: **Call** $D$ threads in each block ($1 \leq j \leq D$)
6: In each thread:
7: qr, x = 0 (on-thread register)
8: r[1 to $D/L$] = 0 (on-thread register)
9: so[1 to $D$] = 0 (shared memory)
10: sg[1 to $D$] = 0 (shared memory)
11: **for** $i \leftarrow N$ **to** 1 **do**
12:     x = 0
13:     qr = q[$bh$][$i$][$j$]
14:     **load** o[$bh$][$i$][$j$] **to** so[$j$]
15:     **load** $\hat{\Omega}$[$bh$][$i$][$j$] **to** sg[$j$]
16:     **for** $m \leftarrow 1$ **to** $D/L$ **do**
17:         r[$m$] += qr×sg[$l \times D/L + m$]×so[$l \times D/L + m$]
18:         x += r[$m$]
19:     **end for**
20:     $\nabla_{\mathbf{k}}\boldsymbol{\Psi}$[bh][i][j] -= x
21: **end for**

---

# B  RNN and Transformer-Based Linear Attention

The original softmax-based attention mechanism in Transformers can be expressed as below, where the $q, k, v$ represent the Query, Key, Value matrices, and $o$ the attention layer output.(See Section 2)

$$\mathbf{o_{ij}} = \frac{\sum_{n=1}^{N} \exp(\mathbf{q}_i.\mathbf{k_n}/\sqrt{D})\mathbf{v_{n,j}}}{\sum_{n=1}^{N} \exp(\mathbf{q}_i.\mathbf{k}_n/\sqrt{D})} \tag{25}$$

By substituting the exponential in Equation 25 with a linear approximation we arrive at Linear Attention (LA), expressed as below

$$\mathbf{o_{ij}} = \frac{\sum_{n=1}^{N} (\mathbf{q}_i.\mathbf{k_n} + 1)\mathbf{v_{n,j}}}{\sum_{n=1}^{N} (\mathbf{q}_i.\mathbf{k_n} + 1)} \tag{26}$$

There are generally two approaches to calculate LA: 1. Transformer-based and 2. RNN-based.

## B.1  RNN-Based

First proposed in 2020 Katharopoulos et al. (2020), Linear Attention (LA) can be implemented using Recurrent Neural Networks (RNNs). Specifically, by defining the hidden state as $S_t = \sum_{i=1}^{t} \mathbf{k}_i.\mathbf{v}_i$ and the update state as $z_t = \sum_{i=1}^{t} \mathbf{k}_i$, Equation 26 can be expressed in the following recurrent format:

$$S_t = S_{t-1} + \mathbf{k}_t.\mathbf{v}_t, \quad \mathbf{o}_t = \frac{\mathbf{q}_t\, S_t}{\mathbf{q}_t\, z_t}. \tag{27}$$

However, the normalizing term, $\mathbf{q}_t z_t$, is observed to cause instability and is often omitted, resulting in $\mathbf{o}_t = \mathbf{q}_t\, S_t$. Recent works have studied use of gates ($\phi(.)$) alternative to the linear kernel, with the general formulation expressed as below

$$S_t = \sum_{i=1}^{t} \phi(\mathbf{k}_i.\mathbf{v}_i), \quad S_t = S_{t-1} + \phi(\mathbf{k}_t.\mathbf{v}_t), \quad o_t = \phi(\mathbf{q}_t)\, S_t. \tag{28}$$

Table 3 provides a summary on recent work and their recurrent formulation.

Table 3: Comparison of RNN-based linear attention formulations.

| Model | Recurrence | Layer Output |
|---|---|---|
| LA Katharopoulos et al. (2020) | $S_t = S_{t-1} + v_t k_t^\top$ | $o_t = S_t q_t$ |
| LA with Normalization | $S_t = S_{t-1} + v_t \phi(k_t)^\top, \quad z_t = z_{t-1} + \phi(k_t)$ | $o_t = q_t S_t/(q_t z_t)$ |
| DeltaNet Schlag et al. (2021) | $S_t = S_{t-1}(I - \beta_t k_t k_t^\top) + \beta_t v_t k_t^\top$ | $o_t = S_t q_t$ |
| Mamba Gu & Dao (2023) | $S_t = S_{t-1} \odot \exp(-(\alpha_t \mathbf{1}^\top) \odot \exp(A)) + (\alpha_t \odot v_t)k_t^\top$ | $o_t = S_t q_t + D \odot v_t$ |
| GLA Yang et al. (2023) | $S_t = S_{t-1} \odot (\mathbf{1}\alpha_t^\top) + v_t k_t^\top, \quad S_t = S_{t-1}\mathrm{Diag}(\alpha_t) + v_t k_t^\top$ | $o_t = S_t q_t$ |
| Mamba-2 Dao & Gu (2024) | $S_t = \gamma S_{t-1} + v_t k_t^\top$ | $o_t = S_t q_t$ |
| Gated DeltaNet Yang et al. (2024) | $S_t = S_{t-1}\left(\alpha_t(I - \beta_t k_t k_t^\top)\right) + \beta_t v_t k_t^\top$ | $o_t = S_t q_t$ |

Traditional RNNs struggle with learning long-term dependencies, which can be addressed with the use of *Forget Gates*. Similarly, Mamba Gu & Dao (2023) introduces a selection mechanism to selectively remember important tokens, and to selectively forget irrelevant ones by choosing to compress or reset the token's state. Furthermore, due to sequential nature of RNNs, RNNs are known to suffer from limited parallelizability. GLA Yang et al. (2023) proposes an efficient GPU implementation chunk-wise attention calculation, where instead of processing the sequence token-by-token (like an RNN) or all at once (which can exhaust memory), the sequence is divided into fixed-size segments, or "chunks". The final hidden state from the end of one

chunk is saved and used as the initial hidden state for the beginning of the next chunk. This "carries over" the contextual information, mimicking the behavior of the sequential RNN and ensuring that information flows across the entire sequence. DeltaNet **??** introduces a novel linear attention mechanism based on the delta rule, a classic error-correction learning principle. Instead of just adding new information to a memory state like in standard linear attention, DeltaNet retrieves an old value from memory associated with the current query, and computes the "delta" or difference between this old value and the new incoming value, and uses this difference to update the memory.

Although the choice of omitting the normalization and use of various gates will result in a considerable deviation from the original Attention mechanism, these models have gained more popularity compared to Transformer-based LA. One reason contributing to this choice is a lack of convenient and efficient implementation of Transformer-based LA.

### B.2  Transformer-Based

The Transformer-based approach follows the same formulation as Equation 26. This implementation closely resembles a standard Transformer architecture, with attention computed using the traditional all-to-all softmax mechanism, but with one key modification: the exponential function is replaced by a linear approximation. This substitution enables reordering of operations, reducing computational complexity from quadratic to linear time (see Section 3.

We should emphasize that the RNN and Transformer-based approaches employ fundamentally different architectures. The RNN-based approach derives layer outputs, $\mathbf{o_i}$, $\forall 1 \leq i \leq N$, sequentially through a recurrent formulation, where each state $i$ depends on the previous state $i - 1$ and a gating function applied to the $i$-th Query, Key, and Value. In contrast, the Transformer-based approach computes outputs by constructing an attention matrix from Query and Key vectors, then multiplying this matrix with the Value vectors. Since the row and column operations in attention matrix construction and matrix multiplication are independent of one another, the computation of all layer outputs $\mathbf{o_i}$ can be performed in parallel for $1 \leq i \leq N$. This makes the Transformer-based approach inherently more parallelizable than its RNN counterpart.

