# OpenReview forum: "Linear Attention Optimized GPU Kernel Implementation"
_TMLR — Rejected by TMLR_

### Review · Reviewer_P4mT · 2025-07-29

**Summary Of Contributions:**

The summary of contributions are as below:

- The authors find a new way of representation the linear attention and its forward and backward passes.
- The authors implemented the CUDA specific kernel for their fourmulation.
- They experimented their results in terms of speed comparison with other CUDA implementations and evaluate the results in language modeling tasks.

**Audience:**

Yes

**Audience Explanation:**

I believe the TMLR audience would be genuinely interested in this paper, as accelerating training and inference in Transformers—particularly linear Transformers—is a highly active and relevant area of research. Recent works such as DeltaNet, MesaNet, and Log-Linear Attention highlight the growing interest in this direction, and this paper contributes meaningfully to that conversation.

**Broader Impact Concerns:**

There is no specific ethical consideration for this study.

**Claims And Evidence:**

Yes

**Claims Explanation:**

The main claims are supported by empirical results and the experimental setup is fair for comparing different methods.

**Requested Changes:**

### Main Concerns

- The main concern with this paper is that it does not properly describe **GLA** or the concept of **linear recurrences**, and in fact, the formulation of GLA presented appears to be incorrect. Most linear Transformer variants, including GLA, apply a **forget gate** to the attention mechanism, which is a crucial component of their effectiveness. This is entirely missing from the paper’s discussion.
I highly recommend that the authors refer to **Table 2 in the DeltaNet paper**, which presents several formulations of linear attention. In particular:
$$
S_i = G_t S_{i-1} + k_i v_i^\top
$$
Here, the gating term \( G_t \) is often **input-dependent**, and the different designs of \( G_t \) result in various model variants. This gating mechanism plays a central role in the performance and expressiveness of modern linear Transformers, yet it is not addressed at all in this paper. The authors should seriously consider incorporating gating into their formulation and analysis.

------

- Additionally, the **title of the paper is misleading**. GLA is not tied to a single model or implementation. **FlashLinearAttention** (https://github.com/fla-org/flash-linear-attention) already support a wide variety of models—including those with gating—and are much more general than what is presented in this paper. The authors should clarify how their proposed implementation compares to these existing, well-known frameworks and justify why they chose not to include or mention **gating** at all.

----

- Finally, the paper does not consider other important components of modern linear Transformers, such as **convolutions**, **extra gating**, or **non-linear activations** applied to queries and keys, which are integral in models like **Mamba**, **DeltaNet**, and **GLA** itself. These omissions limit the relevance and completeness of the paper, and the authors should at least acknowledge and discuss them.

---

### Review · Reviewer_njWe · 2025-09-03

**Summary Of Contributions:**

**Summary**
This work addresses the practical inefficiencies of Linear Attention (LA) in Transformer architectures, despite its theoretical linear time complexity of O(ND2). The authors introduce a novel method for LA’s forward and backward passes based on highly optimized custom CUDA implementations for parallelization and less frequent memory access. It achieves a linear memory complexity of O(ND). The proposed implementation demonstrates substantial performance gains, including much less memory consumption over existing methods, and a huge speedup over FlashAttention-2 in LLM training. It also maintains comparable performance to quadratic attention on major reasoning benchmarks.

**Strengths**
1. Apart from achieving theoretical optimization of O(ND) complexity of linear attention, the work carefully studied the potential real problems of existing linear attention implementations and proposed targeted solutions based on these findings. The work include enough implementation details on how to address these problems.
2. The work conducted extensive experiments to demonstrate the significant improvements on time and memory efficiency of the proposed optimized linear attention implementation.
3. The work also shows that the proposed approach shows similar expressiveness on LLM with billions of parameters compared to traditional attention mechanism while having much less computational footprint.
All these strengths shows that the work is a very solid work.

**Weakness**
The work does not have significant inherent weakness as a work focusing on optimizing the implementation of linear attention mechanism. Please refer to the Requested Changes section for some advice on the work.

**Audience:**

Yes

**Audience Explanation:**

I believe the findings of the paper will be of significant values to the community given the much better scalability of linear attention. The work provide a reusable implementation of linear attention that can be used by research to address other existing problems of linear attention including more iterations required for convergence, gradient decay or explosion.

**Broader Impact Concerns:**

Please refer to requested changes for suggestion on discussions about Broader Impact.

**Claims And Evidence:**

Yes

**Claims Explanation:**

The work did extensive experiments based on its efficient linear attention experiments and compares against both existing linear attentions and efficient traditional attention mechanism implementations like Flash-Attention 2. In all the comparisons, the proposed linear implementation shows superior memory and time efficiency. The work also demonstrates that when scaled to LLM with billions of parameters models based on the linear attention shows similar capacity or performance to models based on quadratic attention while being way more efficient.

**Requested Changes:**

Even though the work is motivated by reducing the memory footprint and computational time, it is worth noting that linear attention can benefit much better from the continuously scaling up of computation resources for modern AI by consuming and processing much longer input sequence given the same amount of memory. I would suggest the work to highlight this benefit of linear attention in the work.

---

> ### Author Response · Authors · 2025-09-18
> **Review Response**
>
> Thank you for your valuable insights. We have added additional content on the benefits of LA in the revised version. We have added a discussion in the Introduction (last paragraph of Section 1). However, we should clarify that the current SOTA softmax-based attention implementations have a memory cost of $O(ND)$ as well (the whole attention matrix is not stored, but rather a number of rows, they then get multiplied with Value, then the next set of rows and so on). The main benefit of LA is the reduced computational complexity.

---

### Review · Reviewer_4eWD · 2025-09-08

**Summary Of Contributions:**

This paper studies an efficient GPU-based implementation of linear attention. Since linear attention has recently attracted interest due to its faster computation compared to standard attention while maintaining comparable empirical performance, an optimized implementation is highly desirable. This paper addresses this issue by proposing a new approach and evaluates its efficiency empirically.

**Audience:**

No

**Audience Explanation:**

Due to the concerns listed above, the current finding of this paper is not valuable for the community.

**Broader Impact Concerns:**

I do not have any concerns.

**Claims And Evidence:**

No

**Claims Explanation:**

The fundamental motivation of this paper, accelerating the computation of linear attention using GPUs, is valuable to the community, which is a strong aspect of this paper.
However, I have several serious concerns regarding the current presentation, evaluation, and quality of the proposed approach.

**Presentation**: The overall writing needs significant improvement. Many parts are redundant, and numerous trivial facts are included unnecessarily. The Introduction is not well structured: since the main motivation of this work is an efficient implementation of linear attention, the background and related work should focus specifically on prior efforts in implementation and optimization techniques, rather than providing only a general overview of attention mechanisms. In addition, much of the mathematical derivation appears trivial or redundant. A substantial rewrite is needed to improve readability and clarity.

**Evaluation**: The empirical evaluation is insufficient to support the claims. In the experiments, the proposed approach is compared against four other methods. However, these methods seem to use different architectures. As a result, it is unclear whether the observed efficiency gains come from the proposed implementation strategy or simply from differences in model architecture. I believe a proper evaluation is needed, where the proposed implementation is compared directly with standard implementations under the same architecture to provide a fair comparison.

**Quality**: It is not clear why the proposed strategy could achieve efficiency gains. From the description, the method seems to decompose matrix multiplication and attempt further parallelization. However, standard matrix multiplication libraries are already highly optimized for GPUs, and intuitively, relying on these existing implementations should yield better performance. A more detailed and convincing discussion is necessary to explain why the proposed method can potentially outperform standard implementations, including an analysis of specific computational bottlenecks and how the proposed strategy addresses them.

**Requested Changes:**

Please address the concerns outlined in the Weaknesses section.

---

> ### Author Response · Authors · 2025-09-18
> **Review Response**
>
> We thank you for your valuable insights. We have addressed the concerns below. Please let us know if they have been sufficiently addressed.
>
> **Presentation:** We have uploaded a revised version of the paper that includes a more detailed explanation of previous work in Appendix B. Regarding the trivial steps, we appreciate the reviewer's observation about these steps. However, the key contribution lies in how these steps are combined and implemented to achieve performance gain compared to existing methods.
>
>
> **Evaluation:** We selected our baselines to ensure a comprehensive evaluation against the current SOTA from multiple angles:
>
>
> * SOTA LA (Over All Architectures): We used Gated-LA, the most efficient LA implementation (which is RNN-based).
>
>
> * SOTA LA (Same Architecture as Ours): We used Speculative Decoding LA, the SOTA LA for the Transformer-based LA, which our method also uses.
>
>
> * Standard Libraries: We compared against PyTorch’s built-in functions to show that our custom CUDA kernels are more performant than highly optimized matrix multiplication libraries (i.e., cuBLAS).
>
>
> * SOTA Softmax Attention: We included FlashAttention-2 as the benchmark for standard, softmax-based attention.
>
> For context, the contributions of prior work have been in kernels for RNNs (Gated-LA) or algorithmic integration (Speculative Decoding LA), which still relies on standard PyTorch functions. Our main contribution is the first efficient CUDA kernel implementation for Transformer-based LA.
> If there is any specific work that we should compare with, which would improve the quality of our evaluation, we would be happy to incorporate it during the revision.
>
>
> **Quality:** The general idea behind our implementation is to come up with a calculation pattern that would allow us to define cuda kernels that achieve a high data reuse rate, and minimize memory access.
> Previous Transformer-based LAs did not develop specific cuda kernels, and are implemented using standard libraries such as Pytorch. One of the reasons standard libraries are inefficient is due to unoptimized memory access for a specific task. For example, to calculate $A×B×C$ using pytorch, the compiler will read A,B from offchip memory, calculate $T=A×B$, store $T$ in offchip, and then access $T,C$ to calculate $T×C$ (aka Eager Mode Execution), with a total of 6 offchip accesses. However, we can write a cuda kernel to access $A,B,C$, calculate $A×B×C$ and then write the result, with a total of 4 offchip accesses. Since each offchip access takes hundreds of cycles, the difference in the execution times of the two implementations is significant.
>
> Regarding the use of matrix multiplication libraries instead of our custom kernels, while it is true that our operations can be expressed as matrix multiplications, this approach would be inefficient. We have provided comparisons against PyTorch's built-in functions by implementing our algorithm in PyTorch rather than using our custom kernels. As shown in Figure 2 and 3, our implementation provides a very significant speedup.\
> To be specific, matrix multiplication libraries are specifically optimized to reduce data movement using patterns tailored for matrix operations (known as 2D tiling). However, this optimization strategy does not align well with our formulation. For example, the factorized values derived in Equation 9 are reused in Equations 8 and 5 to compute the attention layer output. Using standard matrix multiplication would be inefficient because it would require storing intermediate results to off-chip memory and reading them back multiple times, creating substantial data movement overhead.
>
> Regarding the analysis of computational bottlenecks: We have provided the computational complexity for each step in Section 3 (specifically in the final paragraphs of Sections 3.1 and 3.2). Additionally, we analyzed how data movement creates bottlenecks in existing approaches and demonstrated how our method addresses these issues in Figure 4.

---

### Decision · Action_Editor_rAkG · 2025-10-03

**Recommendation:** Reject

**Audience:**

Yes

**Audience Explanation:**

Accelerating training and inference in Transformers is a highly active and relevant area of research.

**Claims And Evidence:**

No

**Claims Explanation:**

The authors have studied inefficiencies of linear attention in transformers and proposed a new method for forward and backward passes through a highly optimized CUDA implementation with controlled memory access. They have compared their implementation with other baselines including FlashAttention-2 for LLM training and showed substantial speedups.

The reviewers have pointed to several strengths in this paper. For example, Reviewer njWe noted “the work carefully studied the potential real problems of existing linear attention implementations and proposed targeted solutions based on these findings. The work includes enough implementation details on how to address these problems”. Reviewer P4mT noted “The main claims are supported by empirical results and the experimental setup is fair for comparing different methods.”

After revisions and discussion with reviewers, all three reviewers raised major concerns that have not been addressed sufficiently during discussions and revisions. In particular, they found that a critical issue with this paper is properly addressing the gated linear attention variants with forget gate. The authors have responded that they focused on Transformer-based approach for LA implementation without any gates. After considering authors' response, the reviewers have not found it satisfactory.

The authors have emphasized in deployment on resource-constrained platforms such as edge devices and mobile hardware while the experiments are on multi-GPU systems with A6000 GPUs that are mainly used for workstations, scientific computing, and numerical simulation.

While I do not recommend acceptance of this paper in its form, I’d encourage the authors to carefully address the reviewers comments and validate efficiency gains on resource-constrained platforms  for an improved revision and future resubmissions.

**Resubmission Of Major Revision:**

The authors may consider submitting a major revision at a later time.